# Bayesian Approach to Disease Risk Evaluation Based on Air Pollution and Weather Conditions

**DOI:** 10.3390/ijerph20021039

**Published:** 2023-01-06

**Authors:** Charlotte Wang, Shu-Ju Lin, Chuhsing Kate Hsiao, Kuo-Chen Lu

**Affiliations:** 1Institute of Epidemiology and Preventive Medicine, National Taiwan University, Taipei 106319, Taiwan; 2Department of Psychiatry, University of California, San Diego, La Jolla, CA 92093, USA; 3Weather Forecast Center, Central Weather Bureau, Taipei 100006, Taiwan

**Keywords:** Bayesian conditional logistic regression, case-crossover study design, air pollutants, meteorological factors

## Abstract

Background: Environmental factors such as meteorological conditions and air pollutants are recognized as important for human health, where mortality and morbidity of certain diseases may be related to abrupt climate change or air pollutant concentration. In the literature, environmental factors have been identified as risk factors for chronic diseases such as ischemic heart disease. However, the likelihood evaluation of the disease occurrence probability due to environmental factors is missing. Method: We defined people aged 51–90 years who were free from ischemic heart disease (ICD9: 410–414) in 1996–2002 as the susceptible group. A Bayesian conditional logistic regression model based on a case-crossover design was utilized to construct a risk information system and applied to data from three databases in Taiwan: air quality variables from the Environmental Protection Administration (EPA), meteorological parameters from the Central Weather Bureau (CWB), and subject information from the National Health Insurance Research Database (NHIRD). Results: People living in different geographic regions in Taiwan were found to have different risk factors; thus, disease risk alert intervals varied in the three regions. Conclusions: Disease risk alert intervals can be a reference for weather bureaus to issue health warnings. With early warnings, susceptible groups can take measures to avoid exacerbation of disease when meteorological conditions and air pollution become hazardous to their health.

## 1. Introduction

Air pollution and weather conditions are known to be associated with respiratory and cardiovascular disease incidence and mortality. Studies have reported the undesirable impact of exposure to air pollutants on human health and the association between disability-adjusted life years and ozone levels [1,2,3]. In a meta-analysis, a positive association between various air pollutants (ozone, particulate matter, NO_2_, SO_2_, CO, etc.) and cardiorespiratory disease has been observed based on studies from twenty-eight countries [4]. This association is not for vulnerable populations such as children and the elderly alone; it is for all age groups and is related to all-cause mortality [5,6,7]. The adverse effect of ozone exposure is usually displayed as a pyramid with severe outcomes such as death or hospital emergency department visits at the top and less severe health conditions for the majority of people at the bottom [3].

In addition to air pollutants, meteorological parameters also play a central role in human health. The findings in a rapidly increasing number of studies in the literature indicate a significant impact of cold and heat on human health outcomes, including morbidity and mortality of cardiovascular diseases [8,9,10,11]. A significant increase in delayed emergency department visits for respiratory and cardiovascular diseases was found to be associated with high temperatures in cities in the US [12,13]. This pattern is consistent with research findings in other countries, e.g., Taiwan, Greece, and India [14,15,16], and the review by Rocque et al. [17]. These meteorological factors affecting health are summarized in Rocque et al. [17] and include high and low temperatures, heatwaves, diurnal temperature range, and humidity. The evidence has accumulated that such environmental stress can lead to acute cardiovascular events or worsen chronic health conditions.

To reduce the deleterious effects of both air pollutants and meteorological parameters on health, the formulation of strategies and responses is needed. Such a formulation may depend on the heterogeneity in susceptible populations, age groups, types of diseases, and differences in the effect sizes of the factors. Studies evaluating these factors and health events in a large population can help develop a statistical model, based on which the degree of heterogeneity and impact can be quantified and a system of recommendations for responses can be designed.

Although previous studies have documented the environmental factors that can impact health conditions, it needs to be clarified how the risk of disease events can be assessed stochastically. The difficulty comes from two sources. First, in addition to the weather conditions and air pollutants, disease is caused by multiple types of individual factors that are relatively heterogeneous. Examples include personal diet, lifestyle, and family history, to name a few. The heterogeneity in these confounding factors creates challenges in data collection and reduces statistical power in analysis. The remedy is either to increase the sample size or to use a matched case-control study so that the heterogeneity can be covered in the large sample or controlled by the matched design [18,19,20,21]. The second challenge is the lack of probabilistic interpretation of traditional frequentist statistical models. The frequentist approach considers the risk (the disease occurrence probability) as a fixed and unknown constant; therefore, evaluating its likelihood, such as the likelihood of the risk being larger than 30%, is not possible. The Bayesian statistic model can be a good solution [18,19].

In this study, we focus on the likelihood estimation of the first attack of ischemic heart disease associated with both air pollutants and weather conditions, to provide a quantified evaluation of the disease occurrence. Specifically, we aim to evaluate probabilistically whether the risk is beyond a certain threshold. Ischemic heart disease, also called coronary artery disease (CAD), has been a major cause of death worldwide. Since this disease is associated with many confounders and risk factors, including high blood pressure, diabetes, smoking, and a sedentary lifestyle, we considered a case-crossover study design so that the confounders can be controlled. Additionally, the case-crossover matching design allows the diseased and non-diseased subjects to be in similar living environments if the weather condition is reasonably stable in the selected time window. We then considered a Bayesian conditional logistic regression model to perform the analysis. This statistical model has received much attention in environmental research [9,18] for its ability to provide a probabilistic estimation and its flexibility in interpretation. The traditional conditional logistic regression can estimate the average risk but cannot estimate how likely the risk is beyond a certain threshold. One advantage of the Bayesian approach with a probabilistic estimation of an individual’s risk [18] is that the Bayesian solution treats the risk as a random quantity rather than a deterministic constant. This can be helpful when, for instance, inferring the chance of an individual’s risk exceeding a threshold or falling within a finite range. The Bayesian approach was adopted in our case-crossover design, where each case subject is matched with the subject’s historical record as the control. This study design has been commonly applied in models of short-term air pollution and acute health events [20,22]. It is similar to the matched case-control study but with only case subjects. This study design can adjust for within-subject bias and control for time-varying confounders, which are both crucial challenges in such environmental research [21,22]. Based on this matching design and the binary (case/control) response variable, the conditional logistic regression model was selected.

This study combined data from three databases in Taiwan: air quality variables from the Environmental Protection Administration (EPA), meteorological parameters from the Central Weather Bureau (CWB), and subject information from the National Health Insurance Research Database (NHIRD). The results of the Bayesian inference are then utilized to construct a risk model for ischemic heart disease as an illustration.

## 2. Materials and Methods

### 2.1. Data Source and Participants

This study collected three types of data: patient, meteorological, and air pollutants. Patient data were collected from the Taiwan National Health Insurance Research Database (NHIRD), which recorded insurance declaration data of patients participating in Taiwan’s National Health Insurance program. Taiwan’s National Health Insurance program was implemented in March 1995, and the coverage rate was around 99.82% at the end of 2018. We defined the susceptible group as those aged 51 to 90 years from 2003 to 2012 who were free from ischemic heart disease (ICD9: 410–414) from 1996–2002. Those patients whose first declared hospitalization was recorded in 2006–2012 were included in the study.

The Central Weather Bureau in Taiwan provided the meteorological data, and we used ground truth data from the FIFOW (Fine Information of Formosa Weather) project in this study. The air pollutant data were downloaded from the Taiwan Air Quality Monitoring Network, established by the Environmental Protection Administration, Executive Yuan (Taiwan). All meteorological and air pollutant data from 2006 to 2012 were used in the data analysis.

### 2.2. Study Design and Statistical Analysis

A case-crossover study design was used to evaluate acute triggers from environmental factors potentially causing disease. In the case-crossover study design, each case serves as his/her own control; that is, the study is self-matched and helpful in evaluating the effects of short-term changes in exposure on transient changes in disease risk. For each patient, we defined the “case window”, the period during which the person was a case, as the week of onset; that is, the week that began six days before the onset of the disease and ended on the day of disease onset. The “control window”, the period during which the person was not a case, was defined as the seven days before the week of onset; that is, the week that lasted from 13 days to seven days before the onset of the disease. The illustration figure is shown in Figure 1. For example, a patient was a case on day 0, and his/her self-matched control was on day −7. A patient with environmental exposure three days before the onset of the disease means that he/she was a case and exposed to environmental factors on day −3, and his/her self-matched control and related exposure in the control period was on day −10.

Before carrying out our Bayesian conditional logistic regression analysis, two steps of data preparation were performed. First, a sine function was used to adjust for the seasonal effect of temperature in the control window. Second, since NHIRD is an insurance declaration database, the database only contains information for insurance declarations but lacks detailed personal information. To link patients’ data with environmental risk factors, we used the method of Lin et al. [23] to estimate where each patient lived. Then, Bayesian conditional logistic regression was used to evaluate the association between environmental factors and the diseases of interest. Let $Y_{it}$ denote the disease status for the *i*-th subject at the *t*-th time point, and $X_{ipt}$ be the *p*-th environmental factor for the *i*-th subject at the *t*-th time point, where *t* = 1 or 2 represents being a case or control, respectively. The complete Bayesian model is formulated as:(1)$$Y_{it}|\pi_{it}\sim Bernoulli\left(\pi_{it}\right)\;\mathrm{where}\;\mathrm{logit}\left(\pi_{it}\right)=\log\left(\frac{Pr\left(Y_{it}\right)}{1-Pr\left(Y_{it}\right)}\right)=\sum_{p=1}^{P}\beta_{p}X_{ipt}\;\beta_{p}\sim Normal\left(0,100\right)$$

The disease status $Y_{it}$ giving the probability $\pi_{it}$ is assumed to follow a Bernoulli distribution where the regression model is applied on the logit scale of this probability, $\mathrm{logit}\left(\pi_{it}\right)$. The regression coefficients $\beta_{p}$ follow a normal prior distribution with zero mean and variance 100. This prior is non-informative and can avoid being subjective. Other non-informative prior can be applied and the results will be similar as long as they are all proper prior. The regression model is described in the equation
(2)$$\mathrm{logit}\left(\pi_{it}\right)=\log\left(\frac{Pr\left(Y_{it}\right)}{1-Pr\left(Y_{it}\right)}\right)=\sum_{p=1}^{P}\beta_{p}X_{ipt}$$

In Bayesian modeling, the posterior distributions of the parameters need to be derived to carry out the inference. Therefore, we performed the MCMC simulations to generate 60,000 posterior samples with a burn-in of 10,000 samples and thinning parameter of 10. Thus, only 5000 posterior samples of each $\beta_{p}$ were used for the analysis and inference. Finally, the personal risk of a given disease based on meteorological and environmental risk factors was calculated by
(3)$$\mathrm{logit}\left(\pi_{i1}\right)-\mathrm{logit}\left(\pi_{i2}\right)=\log\left(odds_{i1}\right)-\log\left(odds_{i2}\right)=\log\left(OR_{i}\right)=\overset{P}{\underset{p=1}{\sum }}\beta_{p}\left(X_{ip1}-X_{ip2}\right).$$

The odds ratio (OR) of a given risk factor is $e^{\beta_{p}}$. According to the classification in Jane et al. (2005), this study design is localizable and ignorable, and therefore the conditional logistic regression yields unbiased estimates.

According to the established Bayesian conditional logistic regression model, we can predict the risk of disease incidence by using the data from weather forecasts and environmental pollution forecasts. Besides indicating the risk of disease incidence, we can also provide disease risk alerts. The disease risk alert intervals are defined by the median and standard deviation of lnOR estimated by substituting the weather and environmental pollution data of the day someone gets a disease. The critical value for each disease risk alert interval is
(4)$$c_{g}=exp\left\{median\left(\ln OR\right)+0.5\times \left(g-2\right)\times SE\left(\ln OR\right)\right\},$$
where g=1 to 6 represents the levels of disease risk alert from low risk to extremely high risk, respectively. Once the OR is predicted and the value falls within a certain disease risk alert interval, a warning can be announced based on this forecast result.

## 3. Results

### 3.1. Bayesian Conditional Logistic Regression

A Bayesian conditional logistic regression model was established to predict the risk of the disease. In this risk prediction model, we considered meteorological factors, air pollutants, seasonal factors, and lag effects as covariates and built separate models for the three different geographic regions. Table 1 shows the posterior means and 95% credible intervals of the regression coefficients based on Bayesian conditional logistic regression to predict the risk of ischemic heart disease. In Northern Taiwan, minimum temperature, maximum temperature, average relative humidity, and maximum relative humidity were associated with ischemic heart disease. In addition to the effects on the day of onset, we also observed lag effects of the environmental factors one day before disease onset and three days before disease onset. In spring and summer, as minimum temperatures on the day of disease onset and three days before onset increased, the risk of the disease decreased (ORs were between $e^{-0.5068}=0.602$ and $e^{-0.1398}=0.870$); that is, cooler weather increased the risk of disease. However, in autumn and winter, minimum temperatures were positively correlated with disease; that is, warmer weather increased the risk of disease. Similar results are shown in Central and Southern Taiwan. In addition, relative humidity in summer and winter was also negatively correlated with the disease risk. The lag effects of relative humidity on disease risk also were observed in all three regions. Unlike in Northern Taiwan and Eastern Taiwan where O_3_ concentration was not correlated with disease, the OR of the average O_3_ concentration on day 0 is $e^{-0.0037}=0.996$ and the O_3_ concentration was negatively correlated with disease in Central and Southern Taiwan; that is, as the average O_3_ concentration increased, the risk of disease decreased.

### 3.2. Risk Prediction and Disease Risk Alert

The critical values were estimated separately based on the three geographic regions’ weather and environmental pollution data. Based on Equation (3) and the coefficients in Table 1, risk prediction models can be derived. For example, the risk prediction model for Northern Taiwan in Spring is
$$\begin{array}{ll} \log\left(OR_{i}\right)= & -0.1614\times \left(\min T{\left(\mathrm{day}\;0\right)}_{i}-\min T{\left(\mathrm{day}-7\right)}_{i}\right)\\ & \;-0.1398\times \left(\min T{\left(\mathrm{day}-3\right)}_{i}-\min T{\left(\mathrm{day}-10\right)}_{i}\right)\\ & \;+0.0109\times \left(\max T{\left(\mathrm{day}\;0\right)}_{i}-\max T{\left(\mathrm{day}-7\right)}_{i}\right)\\ & \;-0.0116\times \left(\max T{\left(\mathrm{day}-1\right)}_{i}-\max T{\left(\mathrm{day}-8\right)}_{i}\right)\\ & \;+0.0074\times \left(\mathrm{ave}\;\mathrm{RH}{\left(\mathrm{day}\;0\right)}_{i}-\mathrm{ave}\;\mathrm{RH}{\left(\mathrm{day}-7\right)}_{i}\right)\\ & \;-0.0090\times \left(\max\;\mathrm{RH}{\left(\mathrm{day}-3\right)}_{i}-\max\;\mathrm{RH}{\left(\mathrm{day}-10\right)}_{i}\right). \end{array}$$

Then, the disease risk alert intervals can be calculated by the median and standard deviation of lnOR estimated by substituting the weather and environmental pollution data of the day someone gets a disease from 2006 to 2012. The critical value for each disease risk alert interval for Northern Taiwan in Spring can be defined via Equation (4). Table 2 shows the medians and standard errors of ORs for three geographic regions in different seasons. Then, the ranges of ORs for each level of disease risk alert for the three geographic regions can be derived based on Table 2 and Equation (4), and the results are shown in Figure 2a–c, respectively. The gradient color from white to dark red represents the levels of disease risk alert from 1 to 6. The median ORs in spring and winter were similar since the weather and environmental conditions in spring and winter are relatively similar in Taiwan. However, the SE of ORs in spring was slightly larger than in winter, so the range of disease risk alert interval in spring was wider than that in winter. It may be because winter gets colder, and there is a larger temperature difference between indoors and outdoors. Once the weather changes significantly, it will increase disease risk. Furthermore, summer and autumn are relatively hot seasons in Taiwan, and the climate changes in autumn are also relatively large. Taiwan is located in a subtropical region, so people are generally less sensitive to hot weather. People in Taiwan are also more tolerant of changes in weather and environmental conditions in summer and autumn. Hence, the median ORs and SEs of ORs in summer and autumn were larger than in spring and winter. That is, the thresholds of disease risk alert in summer and autumn were higher than in spring and winter.

Once the OR is predicted based on the data of weather forecasts and environmental pollution forecasts and the value falls within a specific disease risk alert interval, a warning can be announced. For example, in summer, if the OR for ischemic heart disease in a district in Northern Taiwan is equal to 2.4511 and falls to level 3, an announcement of a level 3 warning would be advised.

## 4. Discussion

The issue of reference time selection in case-crossover studies was discussed in Janes et al. [21]. The authors argued that if there is always a time trend in exposure and reference before the index time, then the estimate may be biased. In this study, neither the pollution data nor the meteorological factors show an increasing or decreasing pattern, whereas temperature obviously does exhibit seasonal trends. Hence, we used a sine function with a period of 365.25 (days) to fit the seven-day temperature and then predicted the seven-day temperature difference. The temperature in the control window was adjusted by this predicted seven-day temperature difference to eliminate bias caused by seasonal changes in temperature. In addition, since the index time distributes across the study timeline and so does the reference time, these two intertwine and hence would be unlikely to exhibit a pattern. Therefore, the assumption of no time trend is considered satisfied in this conditional logistic regression model.

The case-crossover study is useful in evaluating responses that are highly dependent on subject-specific factors, including the individual’s unhealthy behavior, biochemical factors, and family history associated with ischemic heart disease, in this case. Since the NHIRD did not include this detailed personal information and yet it contains a large sample size, the case-crossover design becomes a good choice for controlling variables at the individual level. However, when this information is available, the proposed Bayesian model can be further modified to incorporate such information and achieve better explanatory power.

The use of Bayesian prediction provides a stochastic evaluation of the risk. In other words, the risk of being in different alert intervals can be quantified in terms of probability, which allows individuals or authorities to assess whether the risk is high enough to take action. Such probability depends on the definition of alert intervals constructed based on the critical values determined in Equation (3). These values typically would not be uniform across different areas/countries; they often vary according to the local environment. The approach of Bayesian prediction offers this flexibility and can be applied easily if different scenarios are considered.

Economic assessment is essential for disease risk evaluation, but this issue has not yet been explored in the current study. In the future, it will be possible to cooperate with experts in related fields to explore the economic benefits of using this Bayesian prediction model and disease risk alert intervals for disease prevention to extend the applicability of our study.

In addition, several studies have considered the ventilation coefficient and mixing height as indicators of the dispersion of air pollutants [24,25]. The effect of these two factors on disease risk would be worth pursuing. The current study did not collect this information and thus cannot perform this evaluation. Future studies may take account of these factors and evaluate the degree of influence.

## 5. Conclusions

In this study, we adopted the case-crossover design and used Bayesian conditional logistic regression to build a disease risk prediction model by integrating three databases with patient data, meteorological factors, and air pollutants data. We explored the association between environmental factors and ischemic heart disease. The results show that environmental risk factors are different in different regions, and different seasons. Minimum temperature, maximum temperature, average relative humidity, maximum relative humidity, and average O_3_ are associated with ischemic heart disease. Aside from the effects on the day of onset, we also observed the lag effects of environmental factors. Maximum temperature (day −1) and average relative humidity (day 0) were only significantly associated with ischemic heart disease in Northern Taiwan and Central and Southern Taiwan; maximum relative humidity (day 0) was only significant in Eastern Taiwan, and average O_3_ was only significant in Central and Southern Taiwan. Based on the risk prediction model, we also defined six levels of disease risk alert that could be useful for issuing health warnings. Once the OR is predicted based on the data of weather and environmental pollution forecasts and the value falls within a specific disease risk alert interval, a warning could be released. It is suggested that our findings might be used to establish environmental health risk indicators applied to prevent diseases related to abrupt climate change or air pollutant concentration in public health.

## Figures and Tables

**Figure 1 ijerph-20-01039-f001:**
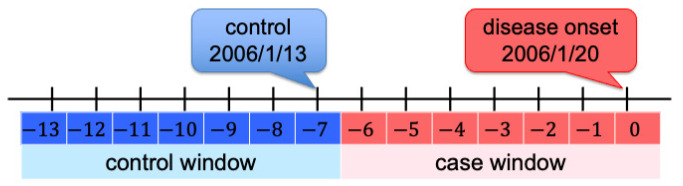
Illustration of selected self-matched samples based on case-crossover study design.

**Figure 2 ijerph-20-01039-f002:**
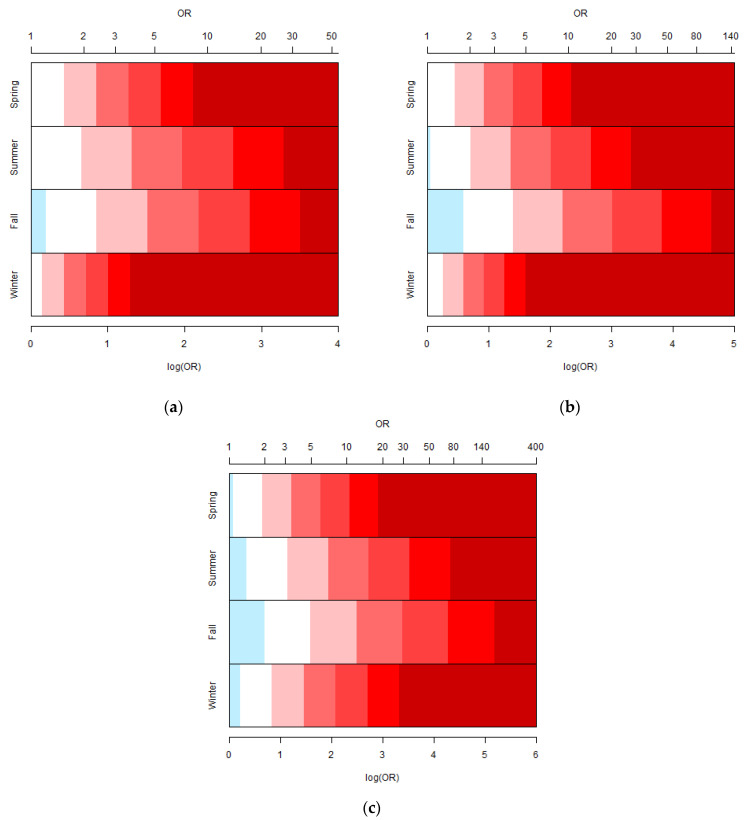
OR ranges for six levels of disease risk alert for ischemic heart disease in each season in (**a**) Northern Taiwan, (**b**) Central and Southern Taiwan, and (**c**) Eastern Taiwan.

**Table 1 ijerph-20-01039-t001:** Posterior mean and 95% credible intervals (CI) of regression coefficients for ischemic heart disease in Northern Taiwan, Central and Southern Taiwan, and Eastern Taiwan.

	Northern Taiwan	Central and Southern Taiwan	Eastern Taiwan
min T ^1^ (day 0) × Spring	−0.1614 (−0.2177, −0.1052)	−0.1900 (−0.2549, −0.1251)	−0.3565 (−0.5351, −0.1779)
min T (day 0) × Summer	−0.2926 (−0.3953, −0.1899)	−0.2565 (−0.3559, −0.1571)	−0.7155 (−1.1067, −0.3243)
min T (day 0) × Autumn	0.4083 (0.3177, 0.4989)	0.4596 (0.3526, 0.5666)	0.5833 (0.3013, 0.8653)
min T (day 0) × Winter	0.1400 (0.0759, 0.2041)	0.0715 (0.0127, 0.1303)	0.1372 (−0.0506, 0.3250)
min T (day −3) × Spring	−0.1398 (−0.1780, −0.1016)	−0.1624 (−0.2171, −0.1077)	−0.2659 (−0.3923, −0.1395)
min T (day −3) × Summer	−0.5068 (−0.6105, −0.4031)	−0.4784 (−0.5825, −0.3743)	−0.5966 (−0.8973, −0.2959)
min T (day −3) × Autumn	0.3747 (0.3063, 0.4431)	0.4635 (0.3698, 0.5572)	0.6308 (0.3768, 0.8848)
min T (day −3) × Winter	0.1011 (0.0629, 0.1393)	0.1557 (0.1081, 0.2033)	0.1883 (0.0607, 0.3159)
max T ^2^ (day 0) × Spring	0.0109 (−0.0259, 0.0477)	−0.0059 (−0.0596, 0.0478)	0.1459 (0.0350, 0.2568)
max T (day 0) × Summer	−0.2289 (−0.3157, −0.1421)	−0.2559 (−0.3457, −0.1661)	−0.1028 (−0.3210, 0.1153)
max T (day 0) × Autumn	0.0319 (−0.0324, 0.0962)	0.1733 (0.0920, 0.2546)	0.0898 (−0.0991, 0.2787)
max T (day 0) × Winter	−0.0192 (−0.0584, 0.0200)	0.0702 (0.0275, 0.1129)	0.2019 (0.0894, 0.3144)
max T (day −1) × Spring	−0.0116 (−0.0398, 0.0166)	0.0147 (−0.0302, 0.0596)	---
max T (day −1) × Summer	−0.0031 (−0.0678, 0.0616)	−0.0856 (−0.1509, −0.0203)	---
max T (day −1) × Autumn	−0.0759 (−0.1316, −0.0202)	0.0356 (−0.0332, 0.1044)	---
max T (day −1) × Winter	−0.0042 (−0.0354, 0.0267)	−0.0194 (−0.0578, 0.0190)	---
ave RH ^3^ (day 0) × Spring	0.0074 (−0.0030, 0.0178)	0.0047 (−0.0110, 0.0204)	---
ave RH (day 0) × Summer	−0.0959 (−0.1210, −0.0708)	−0.0875 (−0.1161, −0.0589)	---
ave RH (day 0) × Autumn	−0.0043 (−0.0190, 0.0104)	−0.0208 (−0.0445, 0.0029)	---
ave RH (day 0) × Winter	−0.0242 (−0.0365, −0.0119)	−0.0300 (−0.0449, −0.0151)	---
max RH ^4^ (day 0) × Spring	---	---	0.0764 (0.0231, 0.1297)
max RH (day 0) × Summer	---	---	−0.1703 (−0.2822, −0.0584)
max RH (day 0) × Autumn	---	---	−0.0520 (−0.1077, 0.0037)
max RH (day 0) × Winter	---	---	−0.0688 (−0.1235, −0.0141)
max RH (day −3) × Spring	−0.0090 (−0.0223, 0.0043)	0.0167 (−0.0015, 0.0349)	0.0353 (−0.0143, 0.0849)
max RH (day −3) × Summer	−0.0605 (−0.0840, −0.0370)	−0.1057 (−0.1378, −0.0736)	−0.065 (−0.1575, 0.0275)
max RH (day −3) × Autumn	0.0119 (−0.0040, 0.0278)	−0.0523 (−0.0786, −0.0260)	−0.0951 (−0.1594, −0.0308)
max RH (day −3) × Winter	−0.0088 (−0.0229, 0.0053)	−0.0374 (−0.0548, −0.0200)	0.0095 (−0.0399, 0.0589)
average O_3_ (day 0)	---	−0.0037 (−0.0068, −0.0006)	---
	(N = 4439, DIC = 5236)	(N = 5195, DIC = 5795)	(N = 693, DIC = 731.1)

^1^ min T: minimum temperature, ^2^ max T: maximum temperature, ^3^ ave RH: average relative humidity, ^4^ max RH: maximum relative humidity.

**Table 2 ijerph-20-01039-t002:** The medians and standard errors of OR were derived from the Bayesian conditional logistic regression model for Northern Taiwan, Central and Southern Taiwan, and Eastern Taiwan in four seasons.

	Northern Taiwan		Central and Southern Taiwan		Eastern Taiwan	
	Median	SE	Median	SE	Median	SE
Spring	1.5329	2.3098	1.5566	2.5703	1.8843	3.1068
Summer	1.9162	3.7221	2.0181	3.6877	3.0837	4.8922
Autumn	2.3354	3.7628	4.0192	5.0102	4.8566	5.9859
Winter	1.1493	1.7685	1.2818	1.9528	2.2779	3.4737

## Data Availability

The air pollutant data were downloaded from the Taiwan Air Quality Monitoring Network at https://airtw.epa.gov.tw/ (accessed on 31 December 2014). Patient data from the NHIRD and meteorological data are not applicable.

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
