# Peer review of "Bayesian Approach to Disease Risk Evaluation Based on Air Pollution and Weather Conditions"

_ijerph, 2023, doi:10.3390/ijerph20021039_

Round 1
Reviewer 1 Report
1. Economic assessment is the essential step for human risk evaluation. Therefore, I propose authors to illustrate this step in the manuscript.
2. In section 3.2, more explanations should be added by authors. Especially, differences in the levels of disease risk alert in accordance with each season should be presented by authors.
3. The effect of weather conditions especially mixing height and ventilation coefficient on disease risk alert in different seasons should be illustrated by authors.
4. In introduction section, more explanations related to the Bayesian conditional logistic regression model should be added by authors to illustrate the novelty of the manuscript.
Author Response
Reply to Reviewer 1
- Economic assessment is the essential step for human risk evaluation. Therefore, I propose authors to illustrate this step in the manuscript.
Reply: We sincerely thank the reviewer for pointing this out. We now state the importance of the economic assessment of risk evaluation in Section 4 and acknowledge the lack of such assessment in our study as a limitation. We plan to collaborate with experts in this area to extend the applicability of our study in the future.
- In section 3.2, more explanations should be added by authors. Especially, differences in the levels of disease risk alert in accordance with each season should be presented by authors.
Reply: In this revision, we modified the first paragraph of Section 3.2 to include more text illustrating the calculation of the levels of disease risk alert, as well as explaining and comparing the differences between seasons in three geographic regions. First, we take a region as an example to illustrate how to calculate predicted OR based on the Bayesian conditional logistic regression model. Then, we added Table 2 to present the medians and standard errors of ORs for three geographic regions in each season and illustrate how to derive the levels of disease risk alert. Finally, we discussed and explained the differences between three geographic regions and four seasons.
- The effect of weather conditions especially mixing height and ventilation coefficient on disease risk alert in different seasons should be illustrated by authors.
Reply: We sincerely thank the reviewer for pointing this out. However, the current study did not collect this information, and thus we cannot perform this evaluation. We now discuss this issue in Section 4 and acknowledge the lack of such information in our study as a limitation. If future research can include relevant information, it is expected that disease risk assessments will be more comprehensive.
- In introduction section, more explanations related to the Bayesian conditional logistic regression model should be added by authors to illustrate the novelty of the manuscript.
Reply: In this revision, we now state with more emphasis on the motivation of this model in Section 1. We first state the aim of performing a likelihood estimation of the disease occurrence probability. We explain why the case-crossover study design is adopted, why conditional logistic regression is used for the matched design, and why the Bayesian approach is considered. The motivation and contribution of this research are now stated in the new fourth and fifth paragraphs in the revision. We hope the revision with the new flow and argument can clarify the goal and novelty of this research. Please see the details in the revised manuscript from Lines 60 to 99 “Although previous studies have documented the environmental factors that can impact health conditions, … the conditional logistic regression model is selected.”
Reviewer 2 Report
In this paper, authors proposed a Bayesian conditional logistic regression model based on a case-crossover design was utilized to construct a risk information system and applied to data from three databases in Taiwan.
But, (i) authors do not explain that the choice of the prior distribution for the beta_p, Is there any other choices?; (ii) In Bayesian conditional logistic regression model, authors don’t explain its motive clearly, Is there any other choices ?; (iii) In Section 2, authors need to explain the assumption and application of Bayesian conditional logistic regression model clearly. Hence, I feel that the main contribution of the paper is weak and the paper cannot be accepted for publication in this applied journal.
Author Response
Reply to Reviewer 2
(i) authors do not explain that the choice of the prior distribution for the beta_p, Is there any other choices?
Reply: In Section 2.2, we now specify the complete Bayesian model, and write out clearly the prior distribution for the regression coefficients beta_p. In addition, we explain using this non-informative prior to remain being objective and we explain the use of other non-informative prior will produce similar results as long as these priors are proper priors.
(ii) In Bayesian conditional logistic regression model, authors don’t explain its motive clearly, Is there any other choices?
Reply: In the revision, the fourth and fifth paragraphs in Section 1 are new and are used to explain the motivation for using the Bayesian conditional logistic regression model and the contribution of this research. Please see the details in the revised manuscript from Lines 60 to 99 “Although previous studies have documented the environmental factors that can impact health conditions, … the conditional logistic regression model is selected.”
(iii) In Section 2, authors need to explain the assumption and application of Bayesian conditional logistic regression model clearly.
Reply: In Section 2.2, we now specify the complete Bayesian model below Figure 1. The explanation and procedures of the Bayesian inference are now stated with more details below the Bayesian model. We also write out the prior distributions of the regression coefficients and explain why using the non-informative priors.
Round 2
Reviewer 1 Report
Dear authors
The manuscript has been well revised.
Reviewer 2 Report
Authors have replied my comments.